

# Inter-comparison of Eddy-Covariance Software for Urban Tall Tower Sites

Changxing Lan[1], Matthias Mauder[1,2], Stavros Stagakis[3], Benjamin Loubet[4], Claudio D'Onofrio[5], Stefan Metzger[6,7,8], David Durden[6], Pedro-Henrique Herig-Coimbra[4]

[1]Institute of Meteorology and Climate Research - Atmospheric Environmental Research (IMK-IFU), Karlsruhe Institute of Technology (KIT), Garmisch-Partenkirchen, 82467, Germany
[2]*Institute of Hydrology and Meteorology, TUD Dresden University of Technology, Dresden, 01069*, Germany
[3]*Department of Environmental Sciences, University of Basel, Basel, 4056, Switzerland*
[4]*ECOSYS, INRAE, AgroParisTech, Universite Paris Saclay, ECOSYS, Palaiseau, 91120, France*
[5]*Department of Physical Geography and Ecosystem Science, Lund University, Lund, 22362, Sweden*
[6]*National Ecological Observatory Network, Battelle, Boulder, CO, 80301, USA*
[7]*Department of Atmospheric and Oceanic Sciences, University of Wisconsin-Madison, Madison, WI, 53706, USA*
[8]*AtmoFacts LLC, Longmont, CO, 80503, USA*

*Correspondence to*: Changxing Lan (changxing.lan@kit.edu)

**Abstract.** Long-term tall-tower eddy-covariance (EC) measurements have been recently established in three European pilot cities as part of the ICOS-Cities project. We conducted a comparison of EC software to ensure a reliable generation of interoperable flux estimates, which is the prerequisite for avoiding methodological biases and improving the comparability of the results. We analyzed datasets covering five months collected from EC tall-tower installations located in urbanized areas of Munich, Zurich, and Paris. Fluxes of sensible heat, latent heat, and $CO_2$ were calculated using three software packages (i.e., TK3, EddyPro, and eddy4) to assess the uncertainty of flux estimations attributed to differences in implemented post-processing schemes. A very good agreement on the mean values and standard deviations was found across all three sites, which can probably be attributed to a uniform instrumentation, data acquisition, and pre-processing. The overall comparison of final flux time-series products showed a good but not yet perfect agreement among three software packages. TK3 and EddyPro both calculated fluxes with low-frequency spectral correction, resulting in better agreement than between TK3 and the eddy4R workflow with disabled low-frequency spectral treatment. These observed flux discrepancies indicate the crucial role of treating low-frequency spectral loss in flux estimation for tall-tower EC systems.

## 1 Introduction

While urban areas cover only a minuscule fraction of the Earth's terrestrial area, approximately 3% as reported by Liu et al. (2014), they are home to more than 55% of the global population, thereby exerting a substantial influence on global greenhouse gas (GHG) emissions (IPCC, 2022). The continued expansion of urban areas is projected to accommodate an estimated 68% (approximately 6.7 billion people) of the world's population by 2050, driven by the ongoing trend of urbanization (UN, 2019). Hence, the pivotal role of urban areas in contributing to global $CO_2$ emissions is widely





acknowledged. This recognition has not only accelerated the development of climate action plans (e.g., China Carbon Neutrality, 2020; C-40, 2022; EU Missions, 2022; Mission net-zero America, 2021) but has also raised growing interest in

existing observation techniques to verify, monitor, and improve estimates of urban $CO_2$ emissions. In addition to satellite observation approaches and modeling frameworks, urban eddy covariance (EC) towers have emerged as a valuable tool for directly monitoring the exchange of $CO_2$ between the land surface and atmosphere with high spatial-temporal resolution (e.g., Vogt et al., 2006; Christen et al., 2011; Järvi et al., 2012; Menzer and McFadden, 2017; Lin et al., 2018; Stagakis et al., 2019). Complementing the ecosystem-focused component of the ICOS network (https://www.icos-

cp.eu/observations/ecosystem) in Europe, more than 15 sites (Table 1), primarily newly established, have been deployed in urban areas (Biraud et al., 2021; Nicolini et al., 2022). Synergies with urban networks, including the US DOE Urban Integrated Field Laboratories (https://ess.science.energy.gov/urban-ifls/) and the Urban Flux Network (https://www.urban-climate.org/resources/the-urban-flux-network/), are established and aim to measure urban emissions and investigate the underlying processes contributing to the diurnal and seasonal patterns of the overall $CO_2$ balance. Within the ICOS-Cities

project (https://www.icos-cp.eu/projects/icos-cities), three additional cities, Munich, Zurich, and Paris are equipped with state-of-art EC measurement instruments.

**Table 1: List of the urban EC towers within the ICOS network (http://www.europe-fluxdata.eu). Tall EC towers established for the ICOS-Cities Project are specified.**

| Location (City, Country) | Measurement Height (m) |
| --- | --- |
| Munich, Germany (ICOS-Cities) | 85.0 |
| Zurich, Switzerland (ICOS-Cities) | 111.8 |
| Paris, France (ICOS-Cities) | 100.0 |
| Berlin, Germany | 56.0 |
| Basel, Switzerland | 39.0 |
| | 41.0 |
| Vienna, Austria | 144.0 |
| Florence, Italy | 33.0 |
| Pesaro, Italy | 23.0 |
| Helsinki, Finland | 31.0 |
| | 45.0 |
| Heraklion, Greece | 27.0 |
| | 24.6 |
| London, United Kingdom | 190.0 |

Compared to the mature ecosystem EC networks, the capacity of tall-tower EC to provide reliable estimates of urban $CO_2$ fluxes remains uncertain due to the paucity of pertinent observations. At the present, there are only a few published examples of tall-tower (e.g., with height reaching the inertial sublayer) urban EC measurements, including London, UK (Helfter et al., 2016); Saika, Japan (Ueyama and Ando, 2016); Beijing, China (Cheng et al., 2018); Vienna, Austria (Matthews and Schume, 2022). Furthermore, the final flux results presented in these studies were derived via either freely distributed software, such



as TK3 and EddyPro, as employed in Cheng et al. (2018) and Matthews and Schume (2022), respectively, or self-developed
processing packages, as in the case of Ueyama and Ando (2016). Although the fundamental principles and assumptions
underpinning the EC technique dictate that the data processing framework (de-spiking, calculation, correction, and data
quality control) should not differ across software packages, variations may arise due to the inclusion of distinct methods, as
extensively discussed in the literature (Mauder et al., 2007). It is noteworthy that even when following identical processing

schemes, different packages might implement them in differing sequences and iterations (Aubinet et al., 2012; Mauder and
Foken, 2006). Consequently, joint efforts have been made to quantify the uncertainties stemming from various data
processing methods and standardize the processing methodology (Aubinet et al., 2012; Lee et al., 2004; Mauder 2007, 2008,
2013; Fratini and Mauder, 2014; Mammarella et al., 2016; Sabbatini et al., 2018). It has been reported that the potential for
deviations in coordinate rotation and detrending methods may account for discrepancies of up to 15% in sensible and latent

heat fluxes, while different high-frequency spectral correction schemes resulted in a 10% discrepancy in $CO_2$ fluxes (Rannik
and Vesala, 1999; Moncrieff et al., 2004; Mauder et al., 2007, 2008). A comprehensive inter-comparison between TK3 and
EddyPro, conducted by developers with in-depth knowledge of the EC method and access to the source code, revealed that
disparities in final fluxes could be minimized through consistent configuration of processing steps and correction schemes
(Fratini and Mauder, 2014). This investigation illuminates that differences in spectral correction schemes were the primary

culprit behind the most significant discrepancies in flux results which proved challenging to eliminate. This software inter-
comparison study highlights the importance of achieving consensus in EC post-processing protocols to ensure robust
comparability across flux measurements.

The culmination of extensive EC software intercomparison studies has significantly contributed to the establishment of a
robust data processing framework for EC data derived from ICOS ecosystem stations (Sabbatini et al., 2018). However, the

75 persistence of uncertainties in flux estimations due to differences in post-processing methodologies remains a pivotal inquiry,
particularly in the context of tall-tower EC measurements in urban areas, which is the main compass of current work. In this
study, we conducted an inter-comparison of friction velocity, sensible heat, latent heat, and $CO_2$ fluxes calculated by three
software packages (i.e., TK3, EddyPro, and eddy4R) at three urban tall-tower EC sites. The primary objective was to
evaluate the influence of different post-processing schemes on the uncertainty of flux estimations. In contrast to TK3 and

80 EddyPro, which are pre-compiled software providing ease-of-use through a graphical user interface with a range of pre-
configured selections, eddy4R is a community-extensible family of R-packages for tower, airborne, and shipborne EC data
processing on the command line, with advanced features like Flux Mapping workflows (Metzger et al., 2017). For
applications other than urban tall towers, eddy4R has previously been compared to TK3 (Metzger et al., 2012) and EddyPro
(Metzger et al., 2017), with excellent agreement in both cases. Notably, eddy4R is used to harmonize data processing across

47 ecosystem EC stations operated by the National Ecological Observatory Network (NEON). For the following
intercomparison, eddy4R is configured based on the NEON workflow in version 1.3.1 (referred to as eddy4R NW hereafter)
with some deviations to facilitate identical data processing for this intercomparison. A range of other workflows and
methods are available, including wavelet-based low-frequency flux inclusion, storage flux and vertical flux divergence.



While such configurations were deemed outside the scope of the current study, they have been used extensively in prior tall

tower and urban research (e.g., Drysdale et al., 2022; Vaughan et al., 2021; Xu et al., 2017, 2018). Here, we focus on a baseline intercomparison of widely accepted turbulence processing schemes as foundation for future work on low-frequency flux inclusion versus low-frequency flux correction, storage flux and vertical flux divergence.

## 2 Datasets. Software, and methodology

As an integral facet of the ICOS-Cities project, new tall-tower EC systems have been established in urbanized areas in three

European cities: Zurich, Munich, and Paris (Figure 1). These systems, featuring uniform instrumentation and employing standardized data acquisition methodologies, are installed either on a telecommunication tower or a meteorological tower situated on the roof of a high-rise building (Figure 2). Specifically, three-dimensional wind velocities, sonic temperature, water vapor, and $CO_2$ concentrations are measured by an IRGASON (Campbell Scientific Inc.), a collocated ultrasonic anemometer, open-path infrared gas analyzer with a 20-Hz sampling frequency. The raw time-series is collected by CR6

datalogger (Campbell Scientific Inc.) and is subsequently streamed to our data server on an hourly basis. This exceptional level of consistency in both instrumentation and data acquisition, a rarity in many other measurement campaigns, allows us to conduct a rigorous investigation for the purpose of conducting the software inter-comparison. It is expected that the outcomes of this study will primarily elucidate differences in methods adopted by different software packages or differences in the implementation of certain methods, emphasizing the importance of this comparative analysis.





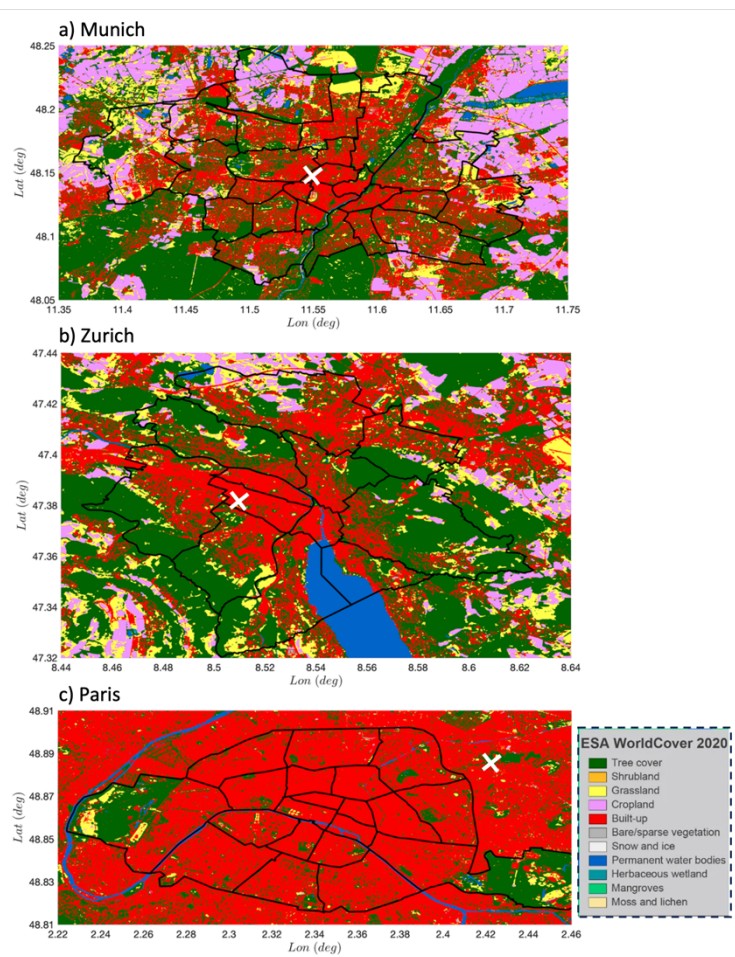

**Figure 1: Land cover map with for the three pilot cities. The land cover map was rendered using the WorldCover product with 10-m resolution provided by European Space Agency (https://esa-worldcover.org). The borders of cities and districts are denoted by thick black lines, while the location of the tall EC tower is illustrated by the white cross.**





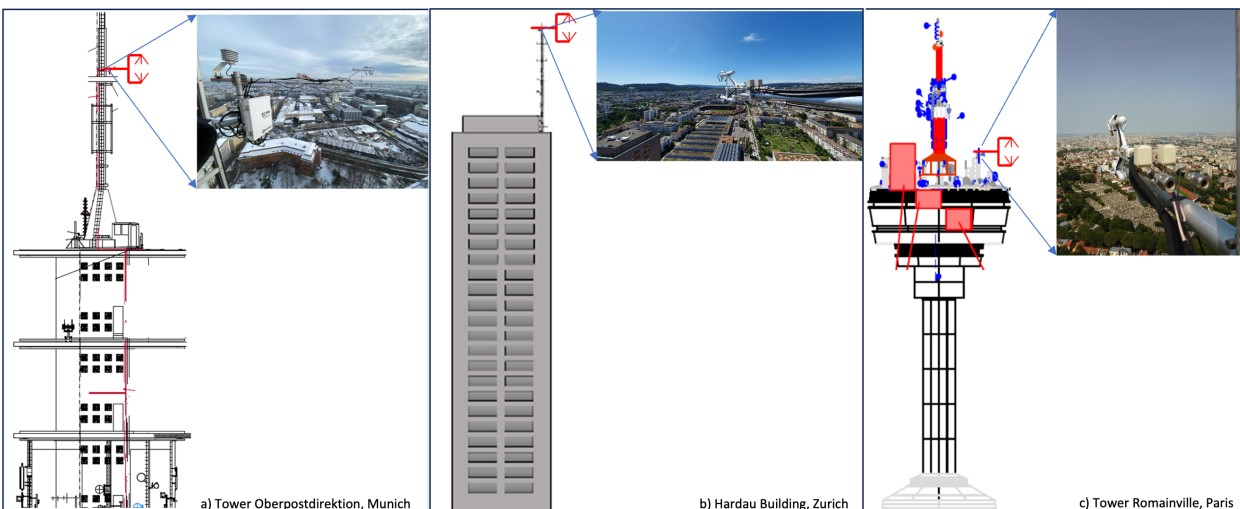

**Figure 2: The schematic of the tower structure and the location of the EC system. The subplots on the top-right are the pictures of the instrumentation taken from the tower.**

Before initiating the computation of fluxes using the three software packages, we subjected the initially measured raw time-series to a time continuity check, which filled missing data points with 'NaN' values. This data preparation ensured that each software package processed complete daily records, thereby guaranteeing that the computed fluxes shared identical timestamps. Given that the three software packages adhered to the same combination of processing steps (Lee et al., 2004), one might anticipate that the final flux outputs would be quite similar. However, distinctions surfaced among the three software packages, not only in terms of the algorithms employed for the de-spiking process but also in their respective flux correction schemes (Figure 3). For instance, the eddy4R NW employs the de-spiking algorithm proposed by Brock (1986) along with an additional threshold recommended by Starkenburg et al. (2016). In contrast, both TK3 and EddyPro adopt Median Absolute Deviation (MAD) for de-spiking (Metzger et al., 2012; Mauder et al., 2013), which is also an option in eddy4R but not selected in adherence to the NEON workflow. While the Webb, Penman, Leuning correction, called WPL (Webb et al., 1980) is used in some eddy4R studies (e.g., Wiesner et al., 2022), it is not incorporated in eddy4R NW because closed-path infrared gas analyzers (e.g., LI-7200, LI-COR Biosciences Inc.) are used at NEON ecosystem stations to measure the dry mole fraction of water vapor and $CO_2$. Indeed, this correction is needed for open-path infrared gas analyzers such as the IRGASON to account for the influence of pressure, temperature, and humidity on density fluctuations, but accounted for in closed-path analyzers through explicit high-frequency ideal gas law conversions. Therefore, to calculate scalar fluxes from mass density of water vapor and $CO_2$ measured by IRGASON, we performed a unit conversion from mass density to dry mole fraction on the raw time-series before initiating the computation (Hartmann et al., 2018). With the advantage of collocation of sonic anemometer and open-path infrared gas analyzer in the IRGASON, this approach is more straightforward and has fewer artifacts compared to performing unit conversion on final fluxes. Significant distinctions also emerge in the spectral loss correction methods implemented by these three software packages. In TK3, the Moore correction



is applied for spectral loss correction in both high-frequency and low-frequency ranges (Moore et al., 1986), while the eddy4R NW corrects only high-frequency spectral loss using a wavelet-based approach, which directly performs correction on the high-frequency time-series rather than on covariances (Nordbo and Katul, 2012). A range of other high-frequency and low-frequency spectral loss treatments are available in eddy4R such as explicit Wavelet inclusion of low-frequency fluxes (e.g., Metzger et al., 2013; Serafimovich et al., 2018; Xu et al., 2018), but not selected for this intercomparison in adherence to eddy4R NW. As for EddyPro, it offers multiple spectral loss correction schemes, but for this study, we adopted the analytical method for both high-frequency (Moncrieff et al., 1997) and low-frequency spectral corrections (Moncrieff et al., 2004), aligning with the processing chain used for EC data measured at ICOS ecosystem sites (Sabbatini et al., 2018).

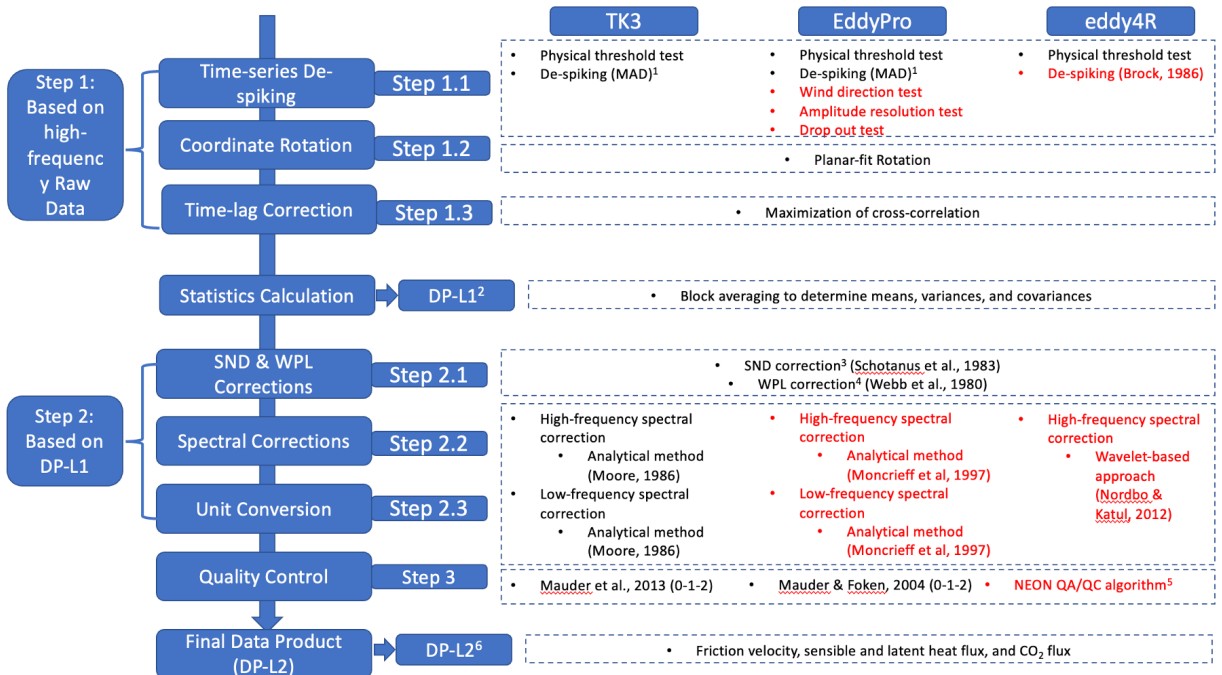

[1] MAD: Median Absolute Deviation (Mauder et al., 2013)
[2] DP-L1: level-one data product including means, variances, and uncorrected covariances
[3] SND correction: conversion of sonic temperature to actual air temperature (Schotanus et al., 1983)
[4] WPL correction: compensating the influence of pressure and temperature on density fluctuations (Webb et al., 1980)
[5]: NEON QA/QC scheme: "-1" indicates the number of spikes within the calculation interval (i.e., 30-min) larger than 10% of the length of the calculation window; "0" indicates final flux results pass both the stationarity and integral turbulence characteristics (ITC) tests; "1" indicates final flux results fail either the stationarity or ITC test
[6] DP-L2: level-two data product including corrected fluxes and the corresponding quality flags

**Figure 3: Processing steps of the EC software packages inter-compared in this work. The overall processing chain aligns with the established protocol for $CO_2$ and energy fluxes calculation at ICOS ecosystem stations. Distinctions in configurations between EddyPro and eddy4R NEON workflow, as compared to TK3, are highlighted in red.**

Our primary focus revolved around the inter-comparison of friction velocity, sensible and latent heat fluxes, and $CO_2$ flux, while statistical values (i.e., mean, standard deviation, and covariance) were also considered to explain the observed discrepancies. Prior to initiating the inter-comparison analysis, the fluxes were subjected to quality screening based on the 0-1-2 quality flag scheme (Mauder et al., 2013). Although the eddy4R NW applies a modular flagging scheme for cross-





discipline integration in place of a traditional rank-based approach (Figure 3), it reports the quantitative results for both the stationarity and integral turbulence characteristic tests. Hence, we utilized the quantitative test results to reassign 0-1-2 data quality flags for fluxes computed by the eddy4R NW, adhering to the methodology outlined by Mauder et al. (2013). Table 2 provides the distribution of final flux results, assigned by the overall quality flags determined through the amalgamated outcomes of the stationarity test and the well-established turbulence test for all three software packages. In addition, TK3

also applies a test for the mean w-offset after planar-fit and interdependency of flags due to corrections or conversions (Mauder et al., 2013). As revealed by prior studies, the residual differences in quality flags were mostly due to different algorithms used for the well-developed turbulence test (Foken et al., 2004; Fratini and Mauder, 2014). However, TK3 tends to classify less data as high quality (i.e., class 0), which can probably be explained by the additional tests described above. It is also interesting to note that data from the Munich site show the largest proportion of high quality data, followed by Zurich

and Paris. These differences can be interpreted as a measure for the suitability of a tower for eddy-covariance measurements. The relatively slim tower structure in the upper 40 meters of the Munich tower probably generates less flow distortion than the more bulky constructions of the towers in Zurich and especially in Paris.

**Table 2: The number of 30-min data segments assigned with different overall quality flags based on the combined results from the stationarity test and well-developed turbulence test calculated by the three software packages.**

|  |  |  | 0 (high quality) | 1 (moderate quality) | 2 (low quality) |
|---|---|---|---|---|---|
| Munich | $u_*$ | TK3 | 3393 | 2992 | 815 |
|  |  | EddyPro | 3611 | 2980 | 609 |
|  |  | eddy4R NW | 3920 | 1644 | 1636 |
|  | $H$ | TK3 | 2068 | 2677 | 2455 |
|  |  | EddyPro | 3949 | 2086 | 1165 |
|  |  | eddy4R NW | 2848 | 2749 | 1603 |
|  | $LE$ | TK3 | 2005 | 2772 | 2423 |
|  |  | EddyPro | 3531 | 2346 | 1323 |
|  |  | eddy4R NW | 2840 | 2313 | 2047 |
|  | $f_{CO_2}$ | TK3 | 2104 | 2379 | 2717 |
|  |  | EddyPro | 4013 | 2013 | 1174 |
|  |  | eddy4R NW | 2853 | 2273 | 2074 |
| Zurich | $u_*$ | TK3 | 1914 | 2604 | 2682 |
|  |  | EddyPro | 2142 | 2773 | 2285 |
|  |  | eddy4R NW | 2018 | 1850 | 3332 |
|  | $H$ | TK3 | 1244 | 1191 | 4765 |
|  |  | EddyPro | 1734 | 2397 | 3069 |
|  |  | eddy4R NW | 1721 | 2816 | 2663 |
|  | $LE$ | TK3 | 1280 | 1228 | 4692 |
|  |  | EddyPro | 1451 | 2484 | 3265 |
|  |  | eddy4R NW | 1711 | 2808 | 2681 |
|  | $f_{CO_2}$ | TK3 | 844 | 1066 | 5290 |
|  |  | EddyPro | 1568 | 2394 | 3238 |
|  |  | eddy4R NW | 1714 | 2808 | 2678 |
| Romainville | $u_*$ | TK3 | 898 | 2701 | 3025 |
|  |  | EddyPro | 946 | 1234 | 3422 |



|  |  |  |  |  |
|---|---|---|---|---|
|  | eddy4R NW | 1043 | 1766 | 3815 |
|  | TK3 | 376 | 1104 | 5144 |
| $H$ | EddyPro | 618 | 1139 | 3545 |
|  | eddy4R NW | 774 | 2352 | 3498 |
|  | TK3 | 311 | 1152 | 5161 |
| $LE$ | EddyPro | 414 | 1104 | 3784 |
|  | eddy4R NW | 516 | 1883 | 4225 |
|  | TK3 | 340 | 942 | 5342 |
| $f_{CO_2}$ | EddyPro | 506 | 1119 | 3672 |
|  | eddy4R NW | 519 | 2630 | 3475 |

The distribution of tilt angles with respect to wind direction was also examined, with the aim of excluding data segments potentially influenced by the building wake or masking effects (Figure 4). Notably, in contrast to the Munich site, large tilt angles were observed in the Zurich and Paris (i.e., Romainville tower) sites, implying a discernible impact of the surrounding architecture and the tower structure on the wind flow. This is likely attributed to the location of the IRGASON. Unlike the EC system in Munich, which is mounted on the needle-like structure of a telecommunication tower, the systems in the

Zurich and Romainville tower sites are situated either on the rooftop of a building or on the platform of a telecom tower, which features a massive antenna on its southeastern side (Figure 2). To minimize the masking effect and flow distortion caused by buildings, data segments with wind direction falling within $\pm 30^o$ of the sonic orientation or tilt angle larger than $10^o$ were excluded from the analysis (Ward et al., 2022; Mammarella et al., 2016). Furthermore, it was also observed that a substantial portion of fluxes corresponding to large tilt angles were marked with 1 or 2 quality levels (Figure 4), emphasizing

the importance of turbulent stationarity test in flux quality assessment for urban EC towers. To evaluate the agreement between the fluxes computed by two different software packages, we employed the symmetric reduced major axis (RMA) linear regression. Despite TK3 not being able to generate an absolute standard of fluxes, it was designated as the reference considering its extensive validation across multiple studies using diverse datasets (Mauder et al., 2007, 2008; Fratini and Mauder, 2014).




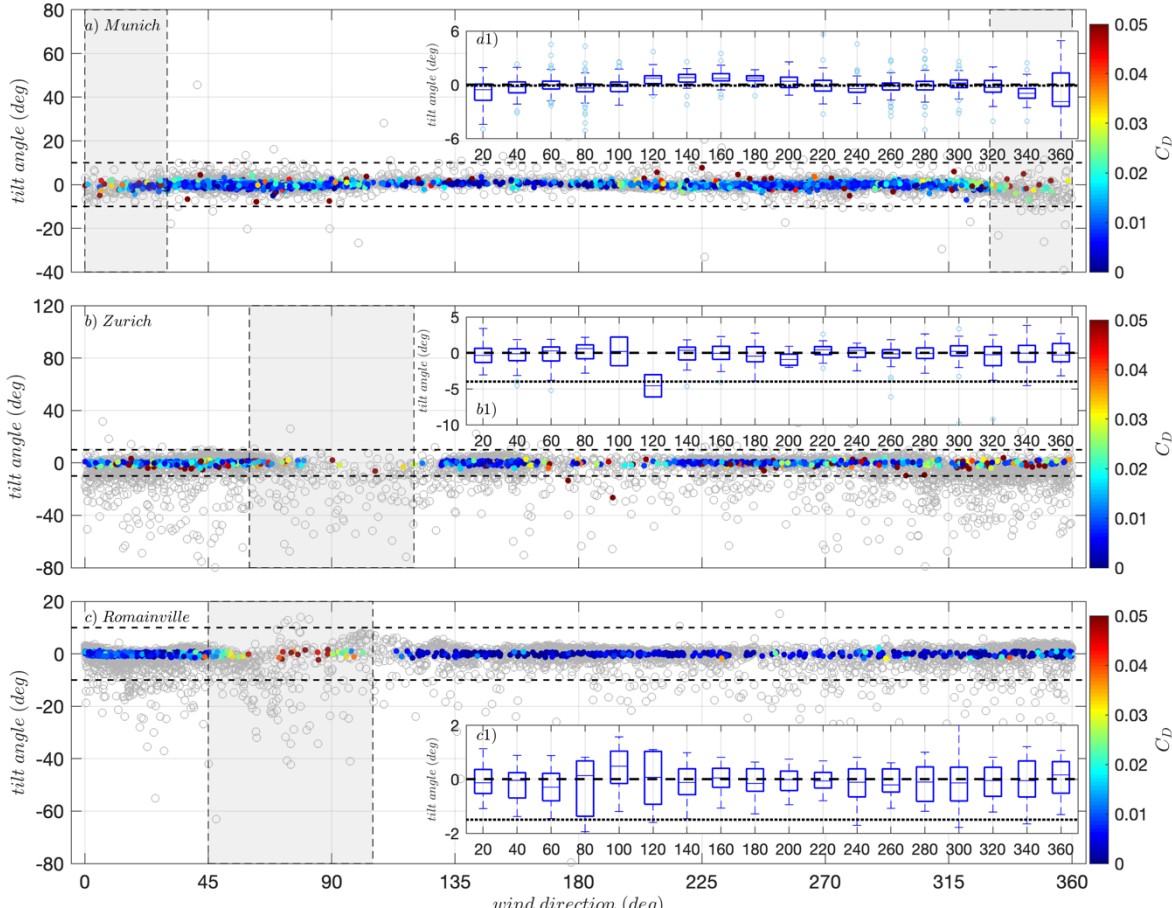

**Figure 4: The distribution of tilt angle $(\frac{\overline{w}}{\overline{u}})$ with respect to wind direction. The gray circles represent all data points before quality flag screening. The solid markers indicate data points assigned a '0' quality flag, with color-coding corresponding to the drag coefficient $(C_D = (\frac{\overline{u}}{u_*})^2)$. The shaded areas denote the wind sectors ruled out due to masking effect. The boxplots (a1 – c1) indicate the median and interquartile of tilt angle as a function of wind sectors only using data points assigned a '0' quality flag. The black dot line indicates the mean value of all data points.**

## 3 Results and discussion

### 3.1 Comparison of mean values, standard deviation, and fluxes

We initiated the analysis by comparing mean values and standard deviations (Figures 5 and 6). The regression statistics revealed a very good agreement across all three sites, which can probably be attributed to the uniformity of instrumentation, data acquisition, and pre-processing (i.e., step 1 in Figure 3) procedures. This finding suggests that differences in de-spiking methods had minimal influence on the derived fluctuation time-series, which were subsequently used to determine covariances. While no systematic differences emerged among the software packages concerning mean values and standard



deviations, some data points related to vertical velocity slightly deviated from the 1-to-1 line. These observed deviations may
be attributed to disparities in the configurations employed to derive planar-fit coefficients in TK3 and EddyPro. In TK3, data

points with horizontal wind speed exceeding $5\ m\ s^{-1}$ were excluded during multiple linear regression, whereas in EddyPro,
outliers were ruled out based on a user-defined threshold for maximum vertical velocity. As evidenced in Figure 7, the
$5\ m\ s^{-1}$ threshold for horizontal wind speed might not be suitable for tall-tower EC systems, as it resulted in the exclusion of
nearly half of the data points when conducting multiple linear regression for determining the planar-fit coefficients. In the
subsequent analysis, therefore, we conducted coordinate rotation in TK3 and eddy4R NW using the planar-fit coefficients

determined by EddyPro to minimize such influence on flux calculations.

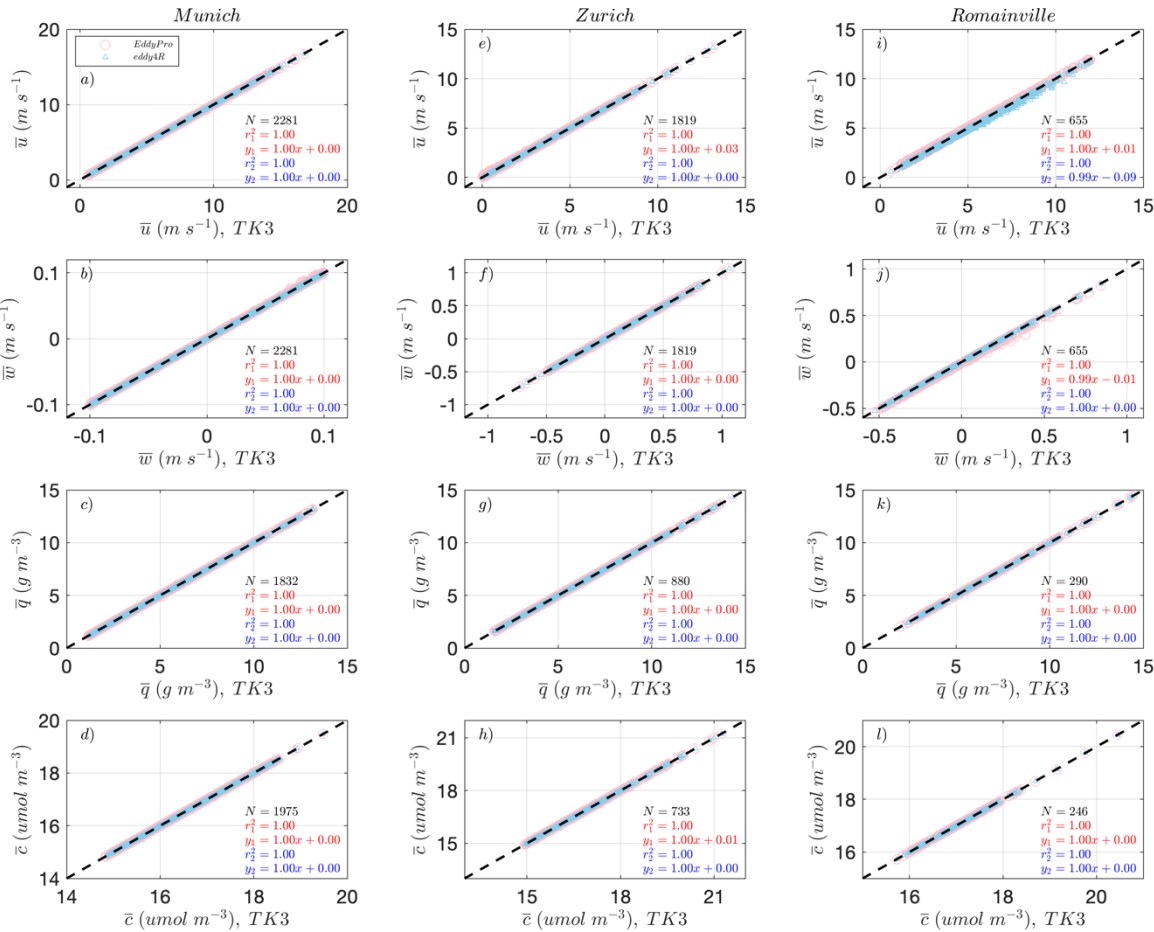

**Figure 5: Comparisons of mean values estimated by the three software packages. The top-to-bottom panels represent the
comparison of horizontal velocity aligned to the streamline (a, e, and i), vertical velocity (b, f, and j), mass density of water vapor
(c, g, and k), and $CO_2$ (d, h, and l). Pink and blue markers denote the comparison between EddyPro and TK3, and eddy4R NW
and TK3, respectively. The black dash line represents the ideal 1-to-1 line. The results of the regression analyses calculated by the
different software packages and the corresponding number of data points are provided in the bottom-right corner of each subplot.**



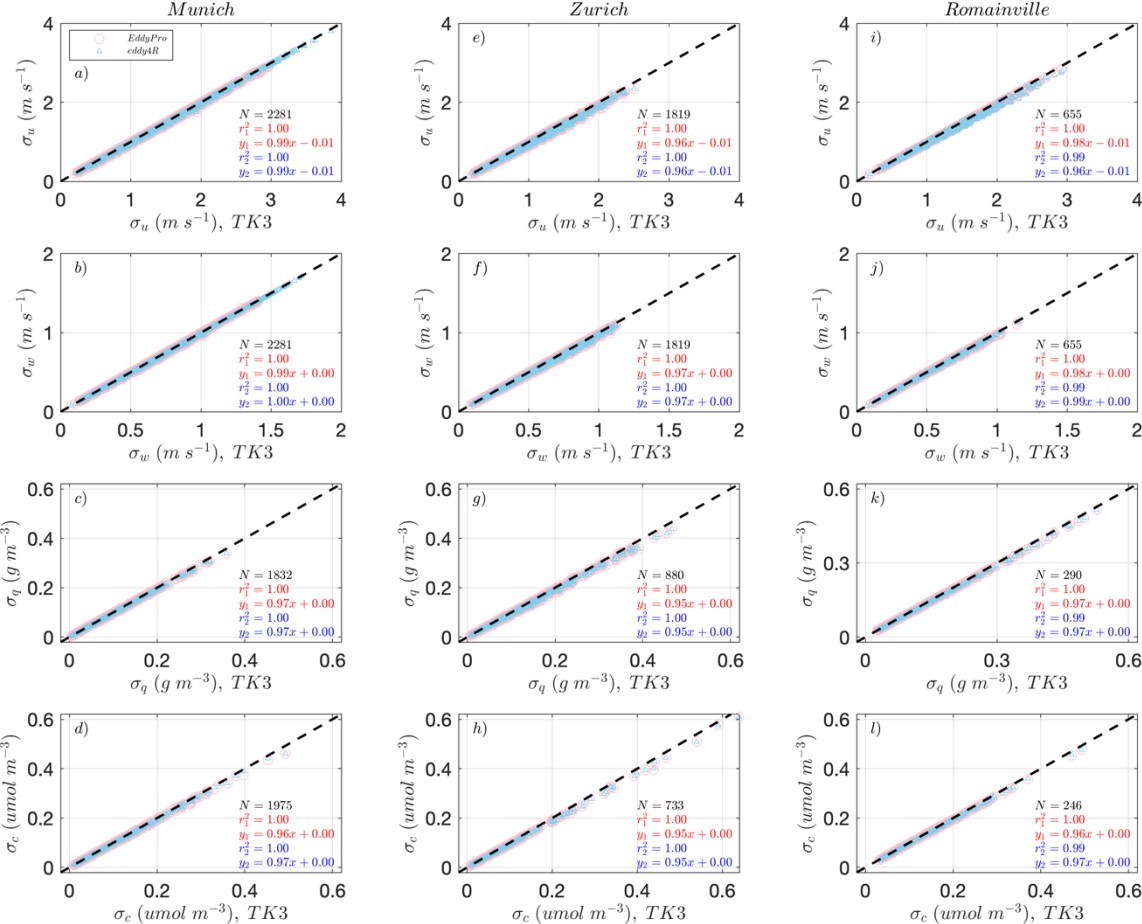

**Figure 6: Comparisons of the standard deviations estimated by the three software packages. The top-to-bottom panels represent the comparison of horizontal velocity aligned to the streamline (a, e, and i), vertical velocity (b, f, and j), mass density of water vapor (c, g, and k), and $CO_2$ (d, h, and l). Pink and blue markers denote the comparison between EddyPro and TK3, and eddy4R NW and TK3, respectively. The black dash line represents the ideal 1-to-1 line. The results of the regression analyses calculated by the different software packages and the corresponding number of data points are provided in the bottom-right corner of each subplot.**





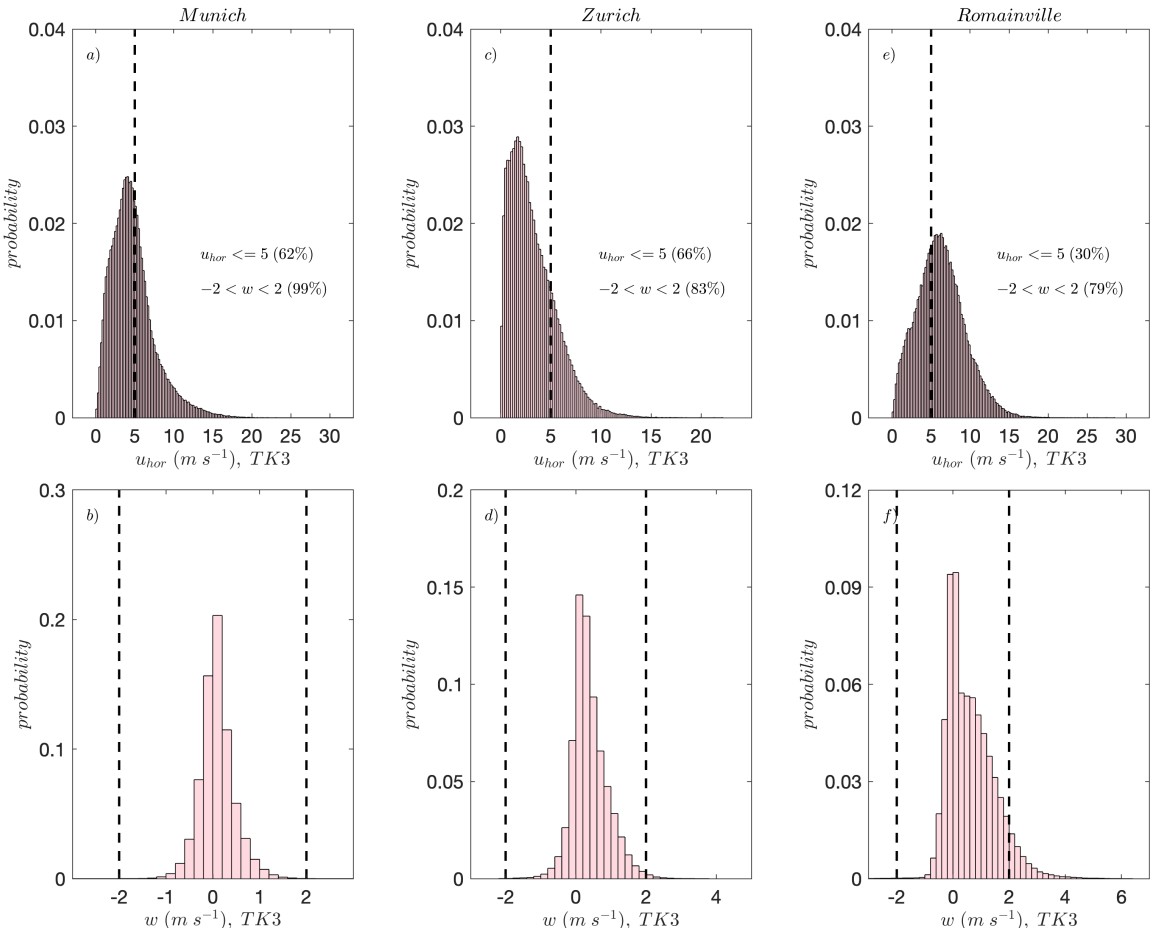

**Figure 7: Histogram of probability density function for originally measured horizontal wind speed (top panels) and vertical velocity (bottom panels). In the top panels, the vertical dashed line represents the threshold of horizontal wind speed configured in TK3, while in the bottom panels, the vertical dashed lines represent the custom-defined range of vertical velocity in EddyPro.**

We proceeded to calculate and compare friction velocity ($u_*$, Figure 8, a, e, and i), sensible heat ($H$, Figure 8, b, f, and j), latent heat ($LE$, Figure 8, c, g, and k), and $CO_2$ fluxes ($f_{CO_2}$, Figure 8, d, h, and l) at each site using the three software packages and the post-processing configurations detailed in Figure 3. Using the identical planar-fit coefficients, the comparison of $u_*$ showed a high degree of concordance, as supported by the $R^2$ values that were near unity. However, a close agreement accompanied by systematic differences in the comparisons of energy and $CO_2$ fluxes were observed. Among these variables, $f_{CO_2}$ showed the most substantial relative bias, consistent with the findings of the prior software inter-comparison study by Fratini and Mauder (2013). Additionally, both the root mean square error (RMSE) and relative bias indicated that fluxes estimated by TK3 and EddyPro were in relatively better agreement than those between TK3 and the eddy4R NW (Table 3). These findings were as expected due to the identical configurations in TK3 and EddyPro, with the exception of the spectral loss correction schemes.





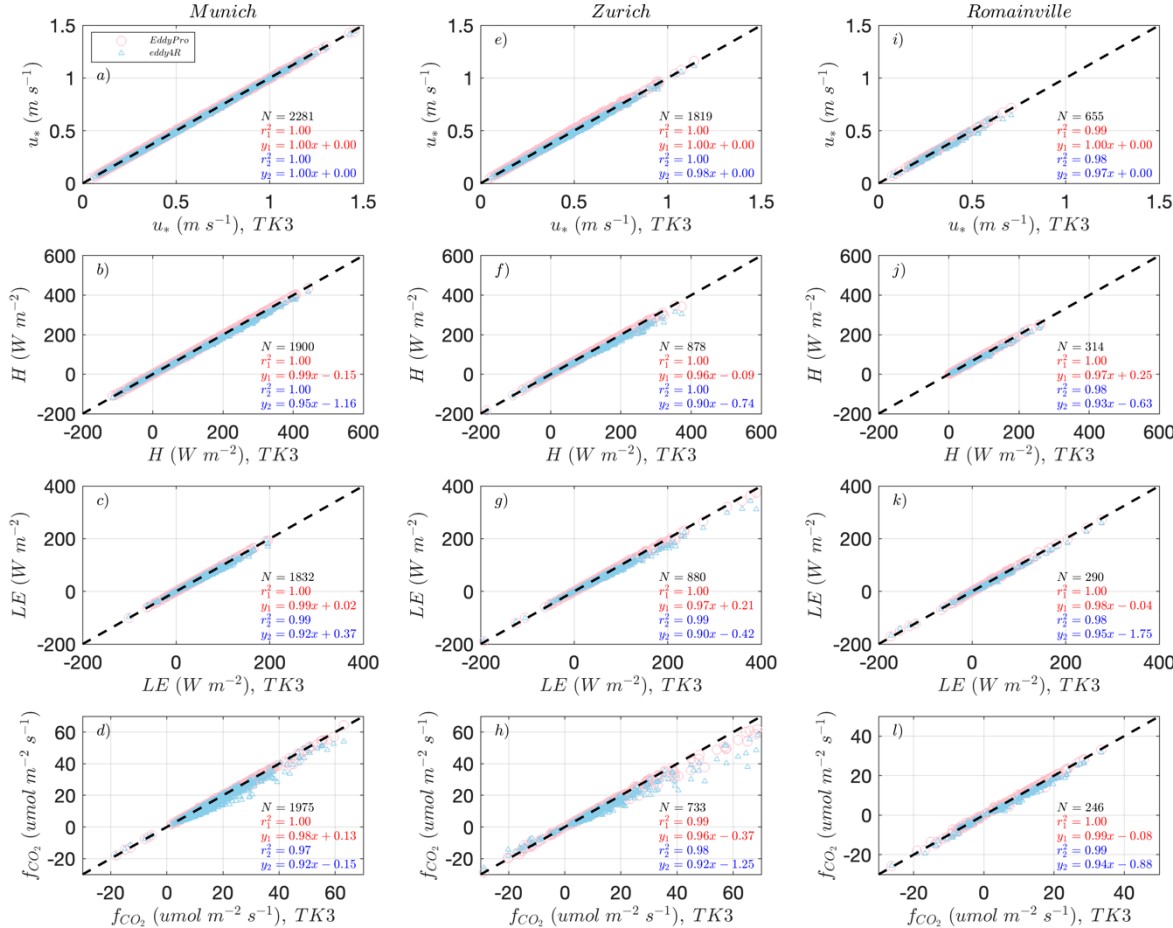

**Figure 8: Comparisons of the final fluxes estimated by the three software packages. The top-to-bottom panels represent the comparison of friction velocity (a, e, and i), sensible heat flux (b, f, and j), latent heat flux (c, g, and k), and CO₂ flux (d, h, and l). Pink and blue markers denote the comparison between EddyPro and TK3, and eddy4R NW and TK3, respectively. The black dash line represents the ideal 1-to-1 line. The results of the regression analyses calculated by the different software packages and the corresponding number of data points are provided in the bottom-right corner of each subplot.**

**Table 3: Summary of the root mean square error and median bias of flux results between two software packages. Note that fluxes computed by TK3 were selected as references.**

|  |  |  | $u_*$ $(m\ s^{-1})$ | $H$ $(W\ m^{-2})$ | $LE$ $(W\ m^{-2})$ | $f_{co_2}$ $(umol\ m^{-2}\ s^{-1})$ |
|---|---|---|---|---|---|---|
| Munich | RMSE | EddyPro | 0.002 | 0.002 | 1.829 | 0.543 |
|  |  | eddy4R NW | 0.008 | 0.009 | 7.030 | 3.898 |
|  | Median Bias | EddyPro | -0.001 | -0.252 | -0.154 | -0.094 |
|  |  | eddy4R NW | -0.005 | -1.629 | -0.905 | -1.020 |





| | | | | | | |
|---|---|---|---|---|---|---|
| Zurich | RMSE | EddyPro | 0.007 | 5.041 | 3.038 | 2.784 |
| | | eddy4R NW | 0.014 | 13.937 | 9.814 | 5.525 |
| | Median Bias | EddyPro | 0.001 | -1.748 | -0.563 | -0.232 |
| | | eddy4R NW | -0.009 | -5.992 | -3.052 | -1.331 |
| Paris | RMSE | EddyPro | 0.002 | 2.310 | 1.925 | 0.564 |
| | | eddy4R NW | 0.023 | 6.358 | 5.832 | 1.749 |
| | Median Bias | EddyPro | 0.002 | -0.560 | -0.307 | -0.171 |
| | | eddy4R NW | -0.010 | -2.880 | -2.660 | -1.138 |

## 3.2 Influence of spectral loss correction on fluxes

Considering that the post-processing (i.e., de-spiking, coordinate rotation, and time-lag correction) done on the raw
time-series had limited impact on the uncorrected covariances, it was reasonable to expect a consistent trend in flux
increments compared to the uncorrected covariance (i.e., Figure 3, covariance in level-1 data product) if the three software
packages employed identical spectral loss correction method. However, as depicted in Figure 9, there was a considerable
variation in the relative differences between final flux results and uncorrected covariance across the three software packages.
This finding confirms that the primary source of the systematic discrepancies observed in flux results (Figure 8) can be
attributed to the different spectral loss correction methods implemented in the three software packages. It is worth noting that
the high-frequency spectral correction method employed by the eddy4R NW generally yielded larger correction values
(order 1%) compared to EddyPro (order 0.1%). A possible advantage of the eddy4R NW wavelet-based spectral correction
method, especially in non-ideal conditions, is that it is not contingent on either a theoretical cospectrum or the cospectral
similarity (Nordbo and Katul, 2012). Another salient feature observed in Figure 9 was the significant increase over the
uncorrected covariances due to the low-frequency spectral loss correction, indicative of substantial flux contributed by large-
scale motions detected by the tall-tower EC systems. Consequently, in contrast to short-tower EC systems, low-frequency
spectral loss correction assumes a more crucial role in correcting fluxes measured by tall-tower EC systems (order 10%).
Hence, the implementation of similar high- and low-frequency spectral loss correction schemes can explain the relatively
small differences in fluxes estimated by TK3 and EddyPro. On the other hand, the disabled low-frequency spectral treatment
in the eddy4R NW can explain the systematic differences in fluxes compared to TK3.



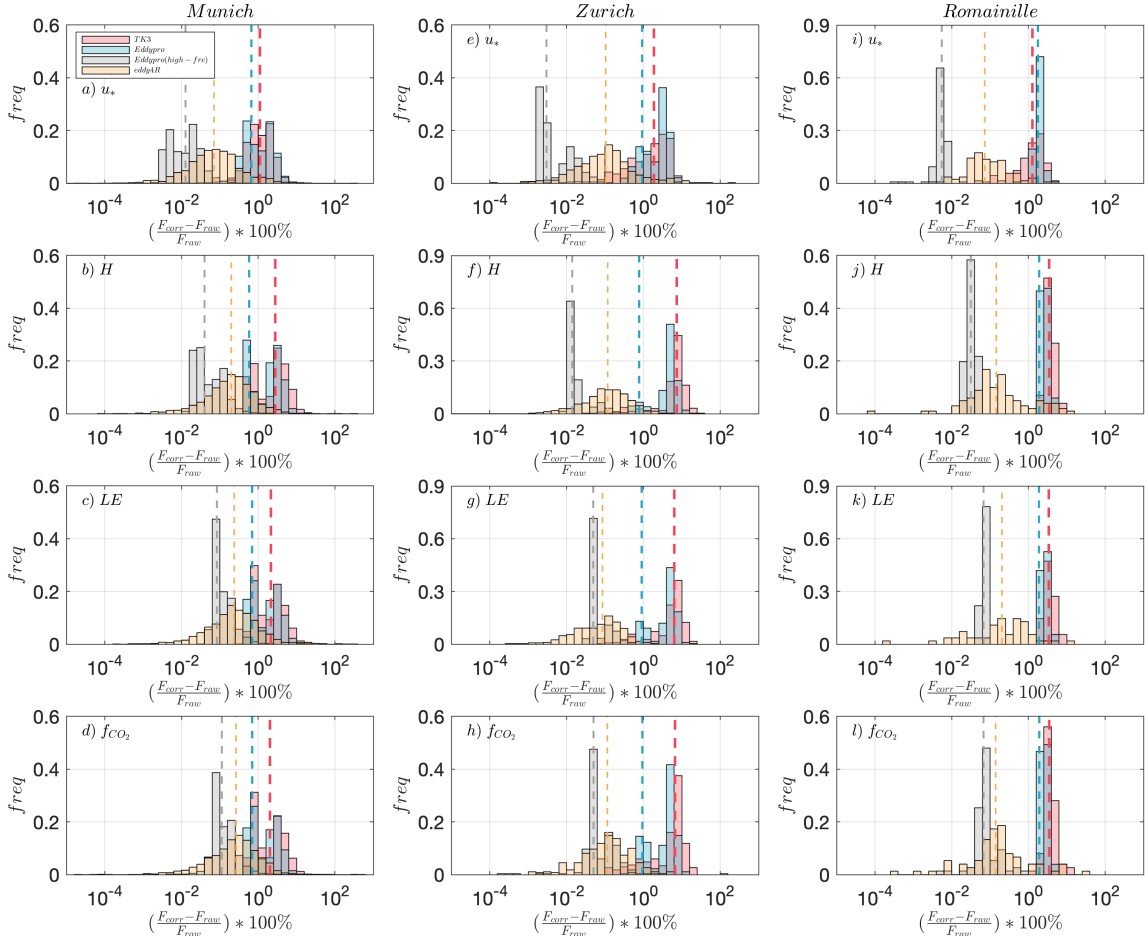

**Figure 9: The frequency distribution of the relative difference between corrected flux and raw covariance. The top-to-bottom panels represent the result of friction velocity (a, e, and i), sensible heat flux (b, f, and j), latent heat flux (c, g, and k), and CO$_2$ flux (d, h, and l). The vertical dashed lines in red, blue, grey, and yellow represent the median values of relative differences corresponding to the results of TK3, EddyPro, EddyPro with only high-frequency spectral loss correction, and eddy4R NW, respectively.**

To further illustrate the systematic discrepancies in fluxes arising from distinct spectral loss correction schemes implemented in the three software packages, we investigated the diurnal pattern of the relative bias between fluxes computed by EddyPro (eddy4R NW) and TK3 (Figure 10). Consistent with features observed in Figure 8, the relative bias of fluxes computed by TK3 and EddyPro did not significantly deviate from the zero line. In contrast, fluxes computed by the eddy4R NW appeared smaller than those calculated by TK3. Notably, the most substantial difference in fluxes calculated by TK3 and eddy4R NW manifested during daytime, indicating a significant increase of daytime fluxes resulting from the low-frequency spectral correction during unstable stratification, similar to the findings from previous inter-comparison between EddyUH and EddyPro (Mammarella et al., 2016). Therefore, we conducted the multi-resolution decomposition (MRD) on scalar fluxes on 4-hour basis to further examine whether the fluxes computed using a 30-min window could capture the contributions from





the large turbulent eddies (Vickers and Mahrt, 2003). As shown in Figure 11, the nighttime MRD cospectra intersected the zero line at a timescale smaller than (or close to) 30 minutes, suggesting that the 30-min averaging period was sufficient to capture the low-frequency flux contributions associated with large-scale motions (Finnigan et al. 2003; Foken et al., 2012). During the daytime, however, the timescales corresponding the MRD cospectrum crossing the zero-line exceeded 30

minutes. This finding indicates that fluxes contributed by turbulent eddies with timescales larger than 30 minutes were not effectively captured, thereby explaining the systematic differences in fluxes computed by TK3 and the eddy4R NEON workflow. This emphasizes the importance of low-frequency spectral loss correction in flux estimation for tall-tower EC systems. Importantly, NEON recognizes the challenge in applying the eddy4R NW originally designed for a median tower height of 22 meters to tall-tower EC systems, and further plans to evaluate the impact of enabling eddy4R low-frequency

spectral treatments for NEON towers and subsequently, compare the fluxes to the counterparts estimated using a longer averaging interval albeit without low-frequency correction as commonly performed at tall towers based on Ogive analysis to determine appropriate averaging intervals. Indeed, eddy4R with low frequency spectral treatment, storage flux, and Flux Mapper enabled has been shown to effectively overcome footprint bias and close the energy balance based on first principles (e.g., Metzger, 2018; Xu et al., 2020).





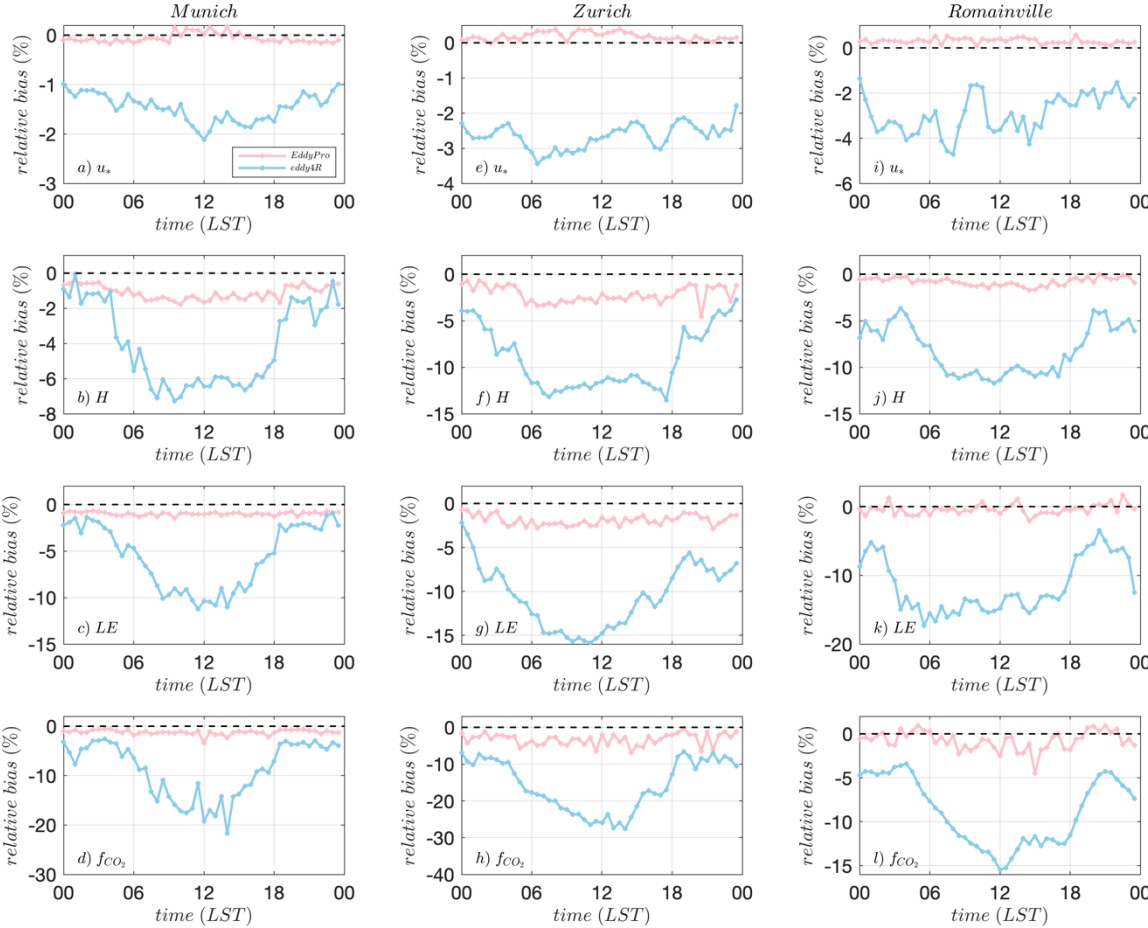

**Figure 10: Median diurnal variation of the relative bias in fluxes. The top-to-bottom panels represent the result of friction velocity (a, e, and i), sensible heat flux (b, f, and j), latent heat flux (c, g, and k), and $CO_2$ flux (d, h, and l). Pink and blue lines denote the relative bias in fluxes between EddyPro and TK3, and eddy4R and TK3, respectively. The horizontal dash line represents the zero line, indicating the estimated fluxes by two software packages are identical.**




**Figure 11: The 4-hour multi-resolution decomposition (MRD) cospectra for fluxes of kinematic heat (top panels), water vapor (middle panels), and $CO_2$ (bottom panels). The pink and blue lines represent the median MRD cospectra for daytime and nighttime, respectively, while the shaded area represents the corresponding interquartile range. The vertical dash line represents the timescale of 30 mins.**

## 4 Conclusions

Through a comprehensive analysis of five months of tall-tower EC measurements across three European pilot cities, we conducted a comparative evaluation of friction velocity, sensible heat, latent heat, and $CO_2$ fluxes computed using three distinct software packages. Our investigation was designed to elucidate the sources of discrepancies in flux estimations caused by different implemented post-processing schemes. Due to the consistency in instrumentation, raw data acquisition, and pre-processing, a very good agreement on the mean values and standard deviations was found. The comparison of the final fluxes showed a remarkable high degree of agreement among the three software packages, especially in comparison to



previous software comparisons, although not yet reaching absolute perfection. The agreement on flux results was largely influenced by the distinctive spectral correction schemes implemented in each software package. Specifically, relative biases in flux estimates between TK3 and EddyPro remained below 1% for $u_*$ and around 2% for scalar fluxes. These minor differences were predominantly caused by different analytical models employed for spectral-loss correction. Conversely, systematic differences in the order of 10% were observed for fluxes estimated by TK3 and the eddy4R NW and primarily attributed to the disabled low-frequency spectral treatment in the eddy4R NW. Our findings emphasized that flux increments resulting from low-frequency spectral-loss correction were an order of magnitude larger than those stemming from high-frequency spectral loss correction. Furthermore, both the diurnal variation in relative flux biases and the MRD cospectra highlighted the crucial role of low-frequency spectral loss correction in flux estimation for tall-tower EC systems. These results constitute a valuable addition to prior software intercomparison studies (Mauder et al., 2008; Fratini and Mauder, 2014; Metzger et al., 2017) by virtue of their unique focus on urban tall-tower EC measurements. Our findings emphasize the significance of a standardized measurement setup and consistent post-processing configurations in minimizing the systematic flux uncertainty resulting from the usage of different software packages. This approach, in turn, ensures the generation of reliable and interoperable flux estimates. Future work evaluating current low-frequency spectral treatment methodologies such as wavelet-based low-frequency inclusion, longer averaging periods, and low-frequency flux correction, as well as storage flux, vertical flux divergence and flux mapping would benefit urban tall-tower EC measurements.

**Code availability**

EddyPro software can be downloaded from the LI-COR Biogeosciences website https://www.licor.com/env/support/EddyPro/software.html. The eddy4R software can be freely accessed at https://github.com/NEONScience/eddy4R. TK3 package can be downloaded from https://zenodo.org/records/20349.

**Data availability**

For the raw 20-Hz eddy covariance data used in this manuscript not available in any repository due to the intensity of the postprocessing and interpretation required. However, the final flux results will be uploaded to ICOS-Cities data portal.

**Author contributions**

C. Lan prepared the initial draft for this manuscript. M. Mauder, S. Stagakis, B. Loubet, C. D'Onofrio, S. Metzger, and Hering-Coimbra participated in the discussion of the intercomparison results and provided valuable contributions to the final manuscript version. M. Mauder develops the TK3 package and provides help in the configuration for flux calculation. S. Metzger and D. Durden provided the eddy4R NEON workflow script.



**Competing interests**

The authors declare that they have no conflict of interest.

**Acknowledgement**

The authors have received funding from ICOS Cities, a.k.a. the Pilot Applications in Urban Landscapes – Towards integrated city observatories for greenhouse gases (PAUL) project, from the European Union's Horizon 2020 research and

335 innovation program under grant agreement no. 101037319. The National Ecological Observatory Network is a program sponsored by the National Science Foundation and operated under the cooperative agreement by Battelle. This material is based in part upon work supported by the National Science Foundation through the NEON Program.

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
