# Peer review of "Inter-comparison of Eddy-Covariance Software for Urban Tall Tower Sites"

_EGUsphere, 2024_

## Author Comment (AC1)

**General Response**

We appreciate greatly all the insightful comments provided by the two anonymous referees. In the following, we address all the comments one by one and revised the manuscript accordingly.

**Anonymous Referee #1**

**Comments**

*1. My major comment pertains to the conclusion of this paper. The author suggests that different spectral loss correction methods contribute to varying CO2 fluxes in the calculation. However, the paper does not specify which spectral loss correction method is more accurate. Does the author have any recommendations on the preferred spectral loss correction method and software to use? Can the author propose a more standard way of computing CO2 flux to reduce uncertainties in CO2 flux calculations?*

**Answer:** We thank the referee for this comment, which echo concerns raised by another referee (Comment #2). Our results show that the slight disparities in fluxes estimated by TK3 and EddyPro result from the different analytical models employed for spectral-loss correction. Conversely, relatively large discrepancies in fluxes estimated by TK3 and edyy4R are mainly attributed to the disabled low-frequency spectral loss correction in eddy4R. These findings highlight the critical significance of low-frequency spectral loss correction in flux estimation for tall-tower EC measurements. Nevertheless, we acknowledge the inherent challenge in definitively determining the most accurate low-frequency spectral loss correction scheme for scalar flux estimation. On one hand, the absence of a definitive "golden" reference for field measurements complicates quantitative evaluations of each spectral loss correction scheme's accuracy. On the other hand, the empirical basis of the low-frequency spectral loss correction schemes implemented in TK3 and EddyPro raises concerns regarding their universal applicability in urban environments. To address these challenges, we are developing a scale-resolved method (based on wavelet analysis and ensemble empirical mode decomposition) for explicitly resolving the flux contribution from large eddies under various stability conditions and wind sectors, rather than relying on spectral correction factors estimated from the empirical functions. An artificial dataset is concocted based on embedding perturbations from intermittent turbulence and asymmetric large eddies into the field observations and quantitatively evaluate

the accuracy of the scale-resolved method in flux estimation. We are committed to addressing these complexities comprehensively in this ongoing research and this scale-resolved flux estimation method will be introduced in the next coming up manuscript.

**2. Figure 10 illustrates a significant bias in fluxes between eddy4R and TK2, which is attributed to the unstable stratification during the daytime. Could the author possibly incorporate the PBLH or other dataset to further support this statement?**

**Answer:** As illustrated by the 4-hour multi-resolution decomposition (MRD) cospectra (refer to Figure 11 in the manuscript), our results indicate that the conventional 30-min flux calculation interval is not sufficient to capture the flux contributions associated with large turbulent eddies (e.g., timescales larger than 30 minutes), particularly durig daytime unstable conditions. Consequently, the observed discrepancies in fluxes estimated by TK3 and eddy4R are primarily attributed to the disabled low-frequency spectral treatment in the eddy4R. We concur with the referee regarding the influence of the planetary boundary layer height (PBLH) on turbulence development. However, it's noteworthy that the PBLH estimates derived from scanning wind lidar data are only available for the Zurich site (Figure R1). Our observations reveal the diurnal pattern of the mixing layer, manifested as its development in the morning, peaking in the afternoon due to thermal convection, while exhibiting relatively lower values during nighttime.

**3. Even though the context suggests that the largest deviation is found in vatical velocity, it is hard to discern from Figures 5,6,8. Would it be better to show the distribution of differences instead of scatter plots?**

**Answer:** We agree with the referee that the Q-Q plots (Figures 5,6,8 in the manuscript) for the comparison of fluxes estimated by TK3, EddyPro, and eddy4R provide limited information due to the relatively strong agreement. Therefore, the distribution of differences is provided (refer to Figures R2 to R4 and in the appendix in the revised manuscript). Figures R2 and R3 show that relatively large discrepancies are observed in velocity components estimated by TK3 and EddyPro caused by the disparities in the configurations in deriving planar-fit coefficients, align with the results presented in Figure 7 of the manuscript. Relatively large differences in mean values and standard deviations for scalars are observed in the comparison between TK3 and eddy4R, attributed to variations in de-spiking techniques employed. On the contrary, systematic

differences in the order of 10% are observed in fluxes estimated by TK3 and the eddy4R, primarily attributed to the disabled low-frequency spectral treatment in the eddy4R This finding is consistent with the feature observed in Figure 9 of the manuscript.

**Anonymous Referee #2**

**Comments**

*1. It is important to set the EC instruments in the inertial sublayer for city-scale, it is better to add the normalized height (with the average building height) in Table1 for the tall towers used in this paper. I am also wondering if the observation height will impact the results because the largest scale of the turbulence captured by the EC maybe different.*

**Answer:** We appreciate the reviewer's suggestion regarding the inclusion of normalized measurement heights relative to the urban canopy height, which indeed bolsters the assertion that measurements are conducted within the inertial sub-layer. Accordingly, we have incorporated the normalized measurement heights in ICOS-Cities into Table 1 of the manuscript, as illustrated below (Table R1).

The influence of the observation height on fluxes is evidenced by the presence of a gap region in the 4-hour multi-resolution decomposition (MRD) cospectra (refer to Figure 11 in the manuscript). This gap region, identified as the timescale corresponding to the cospectra crossing the zero-line, is often employed to separate turbulent motions from sub-meso processes (Howell and Mahrt, 1997; Vickers and Mahrt, 2003, 2006; Vercauteren et al., 2016; Haugeneder et al., 2024). It is noteworthy that the gap region not only varies across observation sites but also varies among different fluxes, indicative of the distinct influence of large eddies on resultant fluxes. This finding further confirms the importance of low-frequency spectral loss correction in flux estimation for tall-tower EC systems. Moreover, it raises concerns about the universal applicability of conventional empirical-based correction functions for low-frequency spectral loss in urban environments. Given its connection of this comment with subsequent discussions and Comment #1 from another referee, we address detailed responses in those sections.

*2. 10% difference in fluxes estimation may lead to quite great bias for long-term observations. May the authors give any suggestion on the selection of the postprocess software, or which is the best one for long-term measurements? This may be difficult when using the field observation but can be done with some artificial perfect 'ideal' data.*

**Answer:** We thank the referee for this significant comment, which aligns with similar concerns raised by another referee (e.g., Comment #1). Our findings underscore the critical importance of

low-frequency spectral loss correction in tall-tower EC measurements, particularly due to the significant flux contribution from large eddies, especially noticeable during daytime unstable conditions. However, conventional low-frequency spectral loss correction schemes, relying on reference functions and 30-minute spectra/cospectra, may not universally apply to urban environments. To address this issue, we are developing a scale-resolved flux estimation method based on wavelet analysis and ensemble empirical mode decomposition. This method aims to explicitly resolve flux contribution from large eddies under different stability conditions and wind sectors, departing from the simplistic multiplication of a spectral correction factor. Additionally, we concur with the regarding the challenge of obtaining a definitive "golden" reference for field measurements. In response to this challenge, we concocted an artificial dataset based on embedding perturbations from intermittent turbulence and asymmetric large eddies into the field observations. This artificial dataset will allow the quantitative evaluation of the accuracy of this scale-resolved method in flux estimation. We are currently in the process of this work and look forward to addressing these considerations comprehensively in the next manuscript.

**Table R1:** List of the urban EC towers within the ICOS network (http://www.europe-fluxdata.eu). Tall EC towers established for the ICOS-Cities Project are specified. The normalized measurement height (with urban canopy height, $h_c$) for the tower-EC systems in ICOS-Cities Project is provided.

| Location (City, Country) | Measurement Height (m) |
|---|---|
| Munich, Germany (ICOS-Cities) | 85.0 ($Z_m/h_c = 4.3$) |
| Zurich, Switzerland (ICOS-Cities) | 111.8 ($Z_m/h_c = 8.4$) |
| Paris, France (ICOS-Cities) | 100.0 ($Z_m/h_c = 4.0$) |
| Berlin, Germany | 56.0 |
| Basel, Switzerland | 39.0 |
| | 41.0 |
| Vienna, Austria | 144.0 |
| Florence, Italy | 33.0 |
| Pesaro, Italy | 23.0 |
| Helsinki, Finland | 31.0 |
| | 45.0 |
| Heraklion, Greece | 27.0 |
| | 24.6 |
| London, United Kingdom | 190.0 |

**Figure R1.** The median diurnal cycle of the planetary boundary layer height (PBLH) in Zurich site estimated from profile measurements of a scanning Doppler lidar. The shaded area represents the interquartile range.

[Figure]

**Figure R2.** The distribution of the relative difference in mean values estimated by EddyPro and eddy4R with respect to the counterparts estimated by TK3.

[Figure]

**Figure R3.** The distribution of the relative difference in the standard deviations estimated by EddyPro and eddy4R with respect to the counterparts estimated by TK3.

[Figure]

**Figure R4.** The distribution of the relative difference in the final fluxes estimated by EddyPro and eddy4R with respect to the counterparts estimated by TK3.

[Figure]

**Reference**

Haugeneder, M., Lehning, M., Stiperski, I., Reynolds, D., & Mott, R. (2024). Turbulence in the Strongly Heterogeneous Near-Surface Boundary Layer over Patchy Snow. Boundary-Layer Meteorology, 190(2), 7. https://doi.org/10.1007/s10546-023-00856-4

Howell, J. F., & Mahrt, L. (1997). Multiresolution flux decomposition. Boundary-Layer Meteorology, 83, 117-137. https://doi.org/10.1023/A:1000210427798

Vickers, D., & Mahrt, L. (2003). The cospectral gap and turbulent flux calculations. Journal of atmospheric and oceanic technology, 20(5), 660-672. https://doi.org/10.1175/1520-0426(2003)20<660:TCGATF>2.0.CO;2

Vickers, D., & Mahrt, L. (2006). A solution for flux contamination by mesoscale motions with very weak turbulence. Boundary-layer meteorology, 118, 431-447. https://doi.org/10.1007/s10546-005-9003-y

Vercauteren, N., Mahrt, L., & Klein, R. (2016). Investigation of interactions between scales of motion in the stable boundary layer. Quarterly Journal of the Royal Meteorological Society, 142(699), 2424-2433. https://doi.org/10.1002/qj.2835